

# Correlation dispersion as a measure to better estimate uncertainty of remotely sensed glacier displacements

Bas Altena[1,2], Andreas Kääb[2], and Bert Wouters[1,3]

[1]Institute for Marine and Atmospheric research, Utrecht University, Utrecht, the Netherlands
[2]Department of Geosciences, University of Oslo, Oslo, Norway
[3]Department of Geoscience and Remote Sensing, Delft University of Technology, Delft, the Netherlands

**Correspondence:** bas altena (b.altena@uu.nl)

**Abstract.** In recent years a vast amount of glacier surface velocity data from satellite imagery has emerged based on correlation between repeat images. Thereby, much emphasis has been put on fast processing of large data volumes. The metadata of such measurements are often highly simplified when the measurement precision is lumped into a single number for the whole dataset, although the error budget of image matching is in reality not isotropic and constant over the whole velocity field. The spread of

the correlation peak of individual image offset measurements is dependent on the image content and the non-uniform flow of the ice. Precise dispersion estimates for each individual velocity measurement can be important for inversion of, for instance, rheology, ice thickness and/or bedrock friction. Errors in the velocity data can propagate into derived results in a complex and exaggerating way, making the outcomes very sensitive to velocity noise and errors. Here, we present a computationally fast method to estimate the matching precision of individual displacement measurements from repeat imaging data, focussing on

satellite data. The approach is based upon Gaussian fitting directly on the correlation peak and is formulated as a linear least squares estimation, making its implementation into current pipelines straightforward. The methodology is demonstrated for Sermeq Kujalleq, Greenland, a glacier with regions of strong shear flow and with clearly oriented crevasses, and Malaspina Glacier, Alaska. Directionality within an image seems to be dominant factor influencing the correlation dispersion. In our cases these are crevasses and moraine bands, while a relation to differential flow, such as shear, is less pronounced.

## 1  Introduction

The increased global availability of satellites images has created unprecedented archives of velocity products over glaciers, ice caps (Fahnestock et al., 2016; Millan et al., 2019) and ice sheets (Rosenau et al., 2015; Joughin et al., 2018). These velocity fields have the large potential to enhance our understanding of ice mechanics and glacier dynamics in space and time. Current efforts are mostly focused on the automatic constructing of large scale time-series (Gardner et al., 2018; Altena et al., 2019;

Derkacheva et al., 2020), or the detection of special speed variations, such as seasonal fluctuations or surge dynamics, from a patchwork of velocity products (Greene et al., 2020; Riel et al., 2021). Advances in time-series construction of glacier velocities will likely mature rapidly in the next few years with the new and increasing availability of suitable data. One promising direction of development is to include the measurement precision into the estimation procedure for glacier velocity variations, either through Bayesian inferences (Brinkerhoff and O'Neel, 2017) or generalized least squares (Altena and Kääb, 2017; Riel



et al., 2021). Though, for such approaches estimation of the dispersion of individual image correlations is needed. Dispersion in this context is the magnitude of fluctuation, or the expected variability of the velocity estimate (i.e.: variance $\sigma$). Typically, a constant variance is set for the whole dataset (known as homoscedasticity), as well as an absence of correlation ($\rho$) between observations of different velocity components (Leprince et al., 2007). The dispersion estimation is then based upon sampling statistics, using a region of bare and stable ground if available, to extract a group variance along each axis (Herman et al., 2011; Heid and Kääb, 2012). However such bare ground might not be a correct representation for glacier surfaces, nor for their correlation dispersion estimate, as the image content and in particular the characteristics of image contrast to be matched are typically different between off- and on-glacier areas, and varies in addition across the glacier surface.

In our opinion the assumption of constant variance (homoscedasticity) does not hold, as displacement extraction is based upon pattern matching of small subsets of imagery, where the image content influences the displacement precision. Pattern matching is based upon a similarity metrics between the matched images across its extent. Such an image subset can have texture with a strong directionality, such as crevasses, or the texture in an image subset is distorted due to skewed flow, such as shear (Debella-Gilo and Kääb, 2012). Both effects are common on glaciers but vary across the scene, thus variation in dispersion might occur across a scene as well. Within image matching, similarity between imagery is computed for a multitude of locations within a neighborhood, resulting in a surface of correlation scores for each potential displacement location, and the maximum peak of this surface is typically detected to indicate the most likely image offset. Since neighboring displacement locations have similar appearance and partial overlap, the similarity score captures smearing in the form of elongated spread of the correlation peak, i.e. such a peak is not a sharp spike but rather a smooth top. For distinct directionality in the matched pattern location, the correlation surface gets elongated in the prevailing direction and such effects can thus be used to extract a better formulation of dispersion for that specific matching location and time interval.

In this contribution we demonstrate a fast estimation approach for dispersion characteristics for individual displacement estimates from image matching. These dispersion characteristics are then used to explore the connection between the correlation spread and the processes of shear flow and crevasse orientation. This gives then for instance a better understanding of the image regions where displacement estimates need to be interpreted with caution. Furthermore, our method enables better quantification of error propagation into the remote sensing and derived model products, which can improve inferences about, for instance, strain-rates, glacier depth, bed roughness and rheology.

## 2 Information within the correlation score surface

The backbone of velocity extraction from imaging satellite is image correlation (a.k.a. pattern matching, feature tracking). For a general overview of an image matching pipeline see Appendix A. The implementation of image correlation is done through the use of a subdomain or kernel in one image that is compared against a second image to find the most similar signal within this subdomain. Typically, a matching domain is a two-dimensional space $(i, j)$, where each axis describes one translation. This





leads to a two-dimensional correlation surface of similarity scores ($\Theta_{i,j}$), where the highest score is taken as the candidate for the displacement.


Apart from the displacement information, also other metrics can be extracted from the correlation surface. We interpret these not as metrics for dispersion, but describing other qualitative aspects. For example, we interpret the absolute value of the highest peak as a proxy for confidence, while an indication for validity can be calculated from the ratio between the highest and second highest peak. Similarly, the signal-to-noise ratio is a proxy of uniqueness, but neither of these descriptors give any

information about the matching precision. The just mentioned reliability proxies are typically provided on a point per point basis within glacier velocity products, while an individual dispersion estimate is still lacking.

However, upon close inspection the width and form of the highest peak in a correlation surface changes and depends a lot on the image content. For example surfaces with a preferred orientation, such as crevasses (Fig. 1a). Here the maximum score

is situated on a ridge of similar high scores, as there is a lack of distinguishable features along the direction of the crevasses. In the direction perpendicular to the dominant feature orientation, the correlation peak is sharp with steep flanks. In the direction of the feature orientation, though, the peak is weakly defined and thus uncertain. Such correlation ridges occur abundantly on glaciers, as elongated features such as crevasses, moraines and streaklines populate many glacier surfaces extensively. Paradoxically, it are these features that exhibit high contrast and are persistent over time, and thus represent very important features

for glacier displacement estimation.

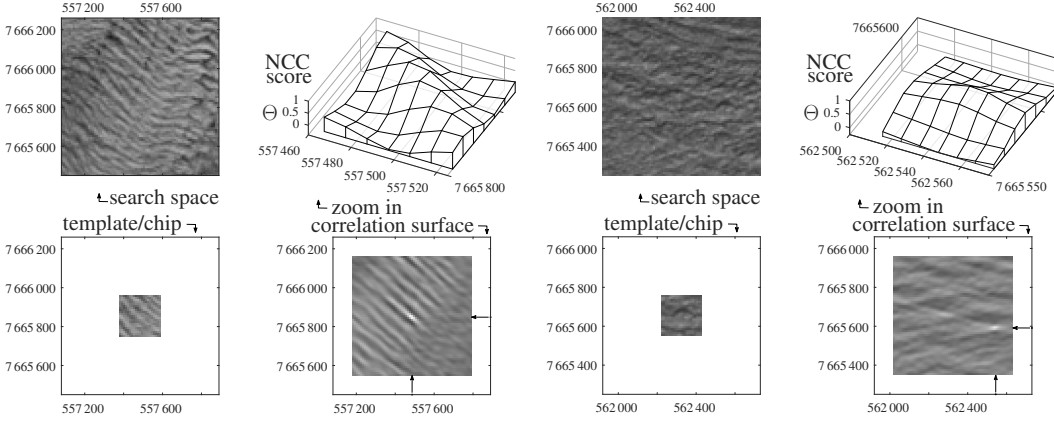

(a) Correlation ridge due to cravesse orientation  (b) Correlation ridge due to shear

**Figure 1.** Examples of cross correlation results with anisotropy, due to the image content or the underlying surface flow. For both panels, lower left panel is the image template to be correlated with the upper left search area. The resulting correlation surface is displayed to the lower right, and a zoom around the correlation peak to the upper right. Both examples (a) and (b) show typical glacier surfaces that result in elongated correlation peaks (here called correlation ridges), sharply defined in one direction, but weakly in the perpendicular direction.





A second process influencing the spread of correlation scores is when significant shear occurs within the template (Fig. 1b). Even though the variation (or contrast) in the template might be present in all directions, a ridge in the correlation surface can emerge similar to the one from crevasses. In case of glacier surface shear, the simple translation assumed in the pattern matching, is not valid (Debella-Gilo and Kääb, 2012). Misalignment in the outer parts of the template, causes dis-similarity, so that the correlation peak gets dampened and neighboring values increase at the same time, weakening the relative strength of the peak. If the size of the template is reduced in such situations, this spreading of correlation scores is reduced, however at the cost of a decreasing signal-to-noise ratio. A second remedy to shear or rotation is to apposition an affine model instead of one with translation only. Though non-linear and thus iterative in nature, such higher order model creates the opportunity to estimate shear and strain rates directly from the image matching (Debella-Gilo and Kääb, 2012).

From the background given in this section, our motivation arises that capturing information about the correlation surface, and in particular its peak bears the potential to better judge the quality and precision of individual matches for displacement measurement.

## 3 Methodology

We perceive the close surrounding of the correlation function as a probability density function. This is a standard perception in the field of fluid mechanics (Bhattacharya et al., 2018), where pattern matching is known as Particle Image Velocimetry (PIV). However, within this latter field of typically controlled laboratory environments image matching is performed on small distinct features, hence shear effects are not present in the templates, while such effects are present for ice flow. Because the correlation surface is perceived as a probability density function it is here fitted with a Gaussian function to be in line with generalized least squares inversion techniques.

### 3.1 Co-variance from correlation spread

Here, we draw up a linear formulation to describe the variance of the correlation peak, which also considers its orientation. At a certain location in this search space $(i, j)$ a two-dimensional Gaussian can be calculated through

$$f(i,j) = A \cdot e^{\left(-\left(a \cdot (i-i_0)^2 + 2b \cdot (i-i_0) \cdot (j-j_0) + c \cdot (j-j_0)^2\right)\right)}. \tag{1}$$

Here $i_0$ and $j_0$ denote the center of the peak ($i_0 = i_{\max} + \Delta i$) which might not coincide with the integer-valued location of the highest value in the correlation grid ($i_{\max}$). The center of the top can be estimated by a peak finding function, and is here considered to be known. Equation 1 is in a simplified form, where $A$ encompasses the magnitude and $a, b, c$ are lumped constants, a detailed derivation thereof is given in Appendix B. Then, the rest of the unknowns can be estimated after some rearrangement,





$$
\underbrace{\begin{bmatrix} \ln\Theta_{i-1,j-1} \\ \ln\Theta_{i-1,j} \\ \vdots \\ \ln\Theta_{i+1,j+1} \end{bmatrix}}_{\mathbf{y}} = \underbrace{\begin{bmatrix} 1 & (i\text{-}1\text{-}i_0)^2 & (i\text{-}1\text{-}i_0)\cdot(j\text{-}1\text{-}j_0) & (j\text{-}1\text{-}j_0)^2 \\ 1 & (i\text{-}1\text{-}i_0)^2 & (i\text{-}1i\text{-}_0)\cdot(j\text{-}j_0) & (j\text{-}j_0)^2 \\ \vdots & \vdots & \vdots \\ 1 & (i+1\text{-}i_0)^2 & (i+1\text{-}i_0)\cdot(j+1\text{-}j_0) & (j+1\text{-}j_0)^2 \end{bmatrix}}_{\mathbf{A}} \underbrace{\begin{bmatrix} \ln A \\ a \\ 2b \\ c \end{bmatrix}}_{\mathbf{x}}.
\tag{2}
$$

This gives the possibility to directly estimate the unknowns (in $\mathbf{x}$) through least squares adjustment from the similarity scores ($\Theta$). The lumped constants $(a,b,c)$ can then be reformulated to extract the variances ($\sigma^2$) and their dependency ($\rho$) from equation 1,

$$
2\rho = \frac{b}{\sqrt{a\cdot c}}, \quad \sigma_i^2 = \frac{1}{-2\cdot(1-\rho^2)\cdot a}, \quad \sigma_j^2 = \frac{1}{-2\cdot(1-\rho^2)\cdot c}.
\tag{3}
$$

This estimation procedure is an extentsion of Anthony and Granick (2009), which only resolved for $\sigma_i$ and $\sigma_j$. However in the formulation of Equation 1 the axes can have a dependency ($\rho$) and correlation ridges with different orientations can thus be estimated. Then the dispersion matrix ($\mathbf{Q}_{yy}$) is composed of the estimates from Equation 3 and the the pixel spacing ($d$) as follows,

$$
\mathbf{Q}_{yy} = \begin{bmatrix} d_x & 0 \\ 0 & d_y \end{bmatrix} \cdot \begin{bmatrix} \sigma_i^2 & \rho\sigma_i\sigma_j \\ \rho\sigma_i\sigma_j & \sigma_j^2 \end{bmatrix}.
\tag{4}
$$

The dispersion matrix ($\mathbf{Q}_{yy}$) can directly be inserted into a covariance matrix for error propagation or data assimulation. The off-diagonal elements of this matrix describe the dependencies between observations. Typically these are set to zero for displacement couples (e.g.: (Derkacheva et al., 2020)) , but they have the ability to describe the temporal and/or spatial relational dependencies within the dataset (a.k.a.: spatial coherency (Riel et al., 2014)).

### 3.2 From (co-)variance to standard error ellipse

For the dependecies between two dimensional displacements, as presented here, interpretation of the elements within the dispersion matrix might not be intuitive. For example, an equal variance can still produce an orientation dependency, as can be seen for example in Fig.2. Hence, here we give the transformation from standard error axis ($\sigma_1^2, \sigma_2^2$), to a description of standard error ellipse in the form of minor and major axis ($\lambda_1, \lambda_2$ respectively) and its orientation ($\rho$). The two axis can be extracted through,

$$
\lambda_1^2 = \frac{\sigma_1^2 + \sigma_2^2}{2} + \sqrt{\frac{(\sigma_1^2 - \sigma_2^2)^2}{4} + \rho\sigma_1\sigma_2}, \quad \lambda_2^2 = \frac{\sigma_1^2 + \sigma_2^2}{2} - \sqrt{\frac{(\sigma_1^2 - \sigma_2^2)^2}{4} + \rho\sigma_1\sigma_2}.
\tag{5}
$$

Similarly, the orientation of the ellipse can be calculated by,





$$\tan(2\theta) = \frac{2\rho\sigma_1\sigma_2}{\sigma_2^2 - \sigma_1^2}. \tag{6}$$

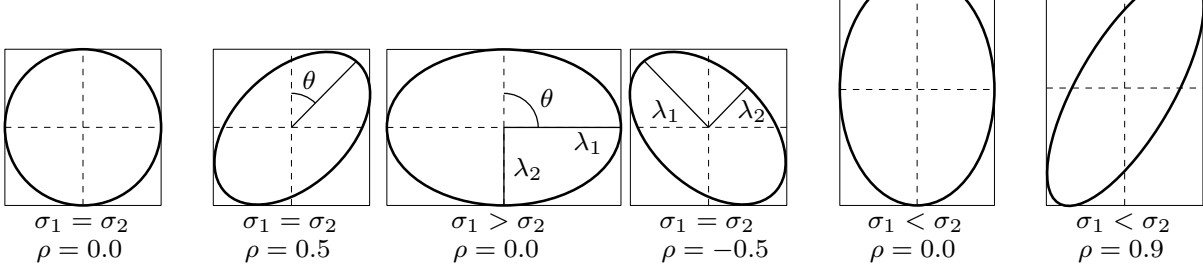

**Figure 2.** Example of ellipses with different dispersion parameters. Illustration adopted from Polman and Salzmann (1996).

### 3.3 Derivatives of flow from incomplete data

Surface strain rates are used in this study to assess the relation between the correlation ridge and ice deformation. Such strain rates can be extracted from a velocity field, however remote sensing results contain wholes and patches without estimates, since similarity could not be established. Hence a robust estimation framework is given in Appendix C, that is somewhat resistant to such sporadic outliers. This procedure is used here to have a more complete strain rate field for analysis.

### 3.4 Crevasse characteristics from Radon transform

To assess the impact of directionality in the input images on our approach to compute and use the dispersion of individual correlations, we need to quantify the directional characteristics of glacier images. In particular crevasse fields have strong directional properties, which can be composed of cracks with several predominant orientations. In order to extract the local crevasse characteristic for each matching template, a Radon transform is used, as described in earlier work (Gong et al., 2018). This methodology provides an argument of the strongest crevasse direction and a strength of this signal. With both the shear flow and crevasse orientation quantified, it is then possible to assess the sensitivity of image matching to these two properties.

## 4 Results

Here we present results from two sites, namely Sermeq Kujalleq, Greenland and Malaspina Glacier, Alaska.



## 4.1 Sermeq Kujalleq, Greenland

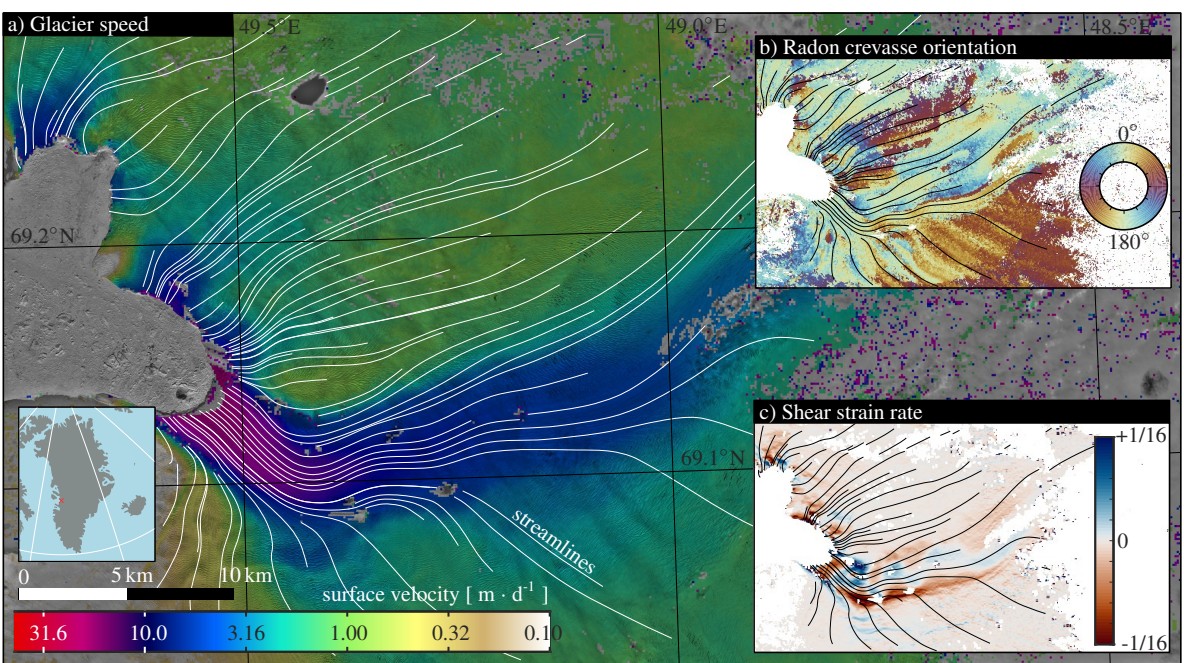

**Figure 3.** Colourcoded speed and streamlines of Sermeq Kujalleq (Jakobshavn Isbræ) between 20th and 30th of July 2020 based on Sentinel-2A imagery. The upper inset (b) shows the dominant crevasse orientation at this time based on the Radon transform, while lower inset (c) shows the shear strain-rate based on this velocity field.

We demonstrate and assess our method to estimate the uncertainty of displacement matching using a small subset of two Sentinel-2A scenes over Sermeq Kujalleq, a large and fast outlet glacier of the Greenland ice sheet. High-pass filtered imagery
(following Fahnestock et al. (2016)) of the 10 meter near-infrared band number 4 are used and the image pair has acquisitions that are ten days apart. We apply a template window of 200 meter in dimension, and velocities are estimated every 100 meter, with a search window of 800 meters. Co-registration is not applied to the image pair beforehand, as related offsets are not the scope of this study, neither does the absence of coregistration influence the outcomes of the presented work. A similar template size of 200 meter was used for the crevasse detection using the Radon transform.

The velocity magnitude between our two images (20 and 30 July, 2020), derived streamlines, and the results of the Radon transform (i.e. the main crevasse orientation) are shown in Fig. 3. The streamlines indicate a strongly convergent flow of this outlet glacier. At some places the signal-to-noise ratio of the image matching was too low (SNR < 4) and the displacements have been excluded. This happened in particular at the Eastern part, where a cloud is present in one of the images, and at other
locations which seem to correspond to supraglacial lakes. The lower right inset of Fig. 3 shows the shear strain-rate. A major feature is a large shear zone along the southern flank of the main flow channel. Finer details like alternating patches are also





present in the main outlet, which could stem from the propagation of subglacial features to the surface. The dominant crevasse orientation (upper right panel in Fig. 3) is transverse to the flow direction, aligning with crevasses originating from extensive flow. Some regions have more complex orientations, most likely due to variations in surface slope and bedrock variation.

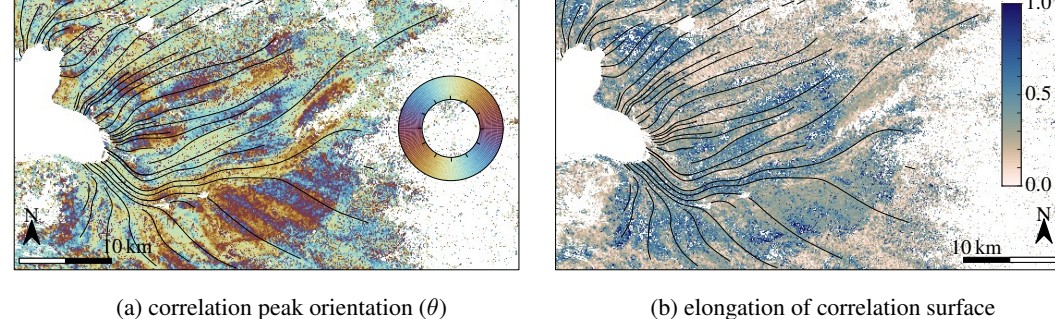

(a) correlation peak orientation ($\theta$)    (b) elongation of correlation surface

**Figure 4.** Correlation peak parameters for individual image matching results, describing the attitude (Fig. 4a, correspond this to Fig. 3b as well) and the strength of its a-symmetry (Fig. 4b).

The orientation of the correlation ridge ($\theta$) for the two example images is shown in Figure 4a, and in general has a stark resemblance with the orientation of the crevasses (Fig. 3b). In Figure 4b a measure for the elongation of the correlation ridge is plotted. Here, elongation is given as the normalized inequality of the two dispersion components ($[\min(\lambda_x, \lambda_y) - \max(\lambda_x, \lambda_y)]/[\lambda_x + \lambda_y]$). Hence 0 corresponds to a perfectly circular distribution, while 1 would be a straight ridge.

## 4.2 Malaspina Glacier, Alaska

Results from the surroundings of Malaspina and Agassiz Glacier in the St Elias Mountains are presented here as well. The region exhibits more glacial features than Sermeq Kujalleq, which is an outlet of Greenland ice sheet with predominantly clean ice. For example, a large collection of morraines, ogives, foliations, meltwater channels and orientations of flow are present on both Malaspina and Agassiz Glacier, as can be seen in Figure D1 in the Appendix. Malaspina Glacier is an outlet of Seward

Glacier with a total area of $5\,000$ km$^2$ (Molnia, 2008), its entire piedmont lobe lies within the ablation area. Agassiz Glacier is the other large tributary of Malaspina Glacier, and creates a distinct Western lobe. The ice transport from Seward glacier has multi annual fluctuations (see youtube-link and (Altena et al., 2019) ), creating looped or curved morrain bands.

Here we use two subsets of Sentinel-2 scenes from the 21$^{st}$ of August and the 15$^{th}$ of September 2019, a 25 days differ-

180 ence. Processing parameters are similar to the Sermeq Kujalleq study: a high-pass filtered band 4 image was matched, with a template window of 200 meter wide, and velocities are estimated every 100 meter, with a search window of 800 meters. No co-registation over stable ground was done, so the velocities should be seen as displacement (being real surface displacementes or artificial).





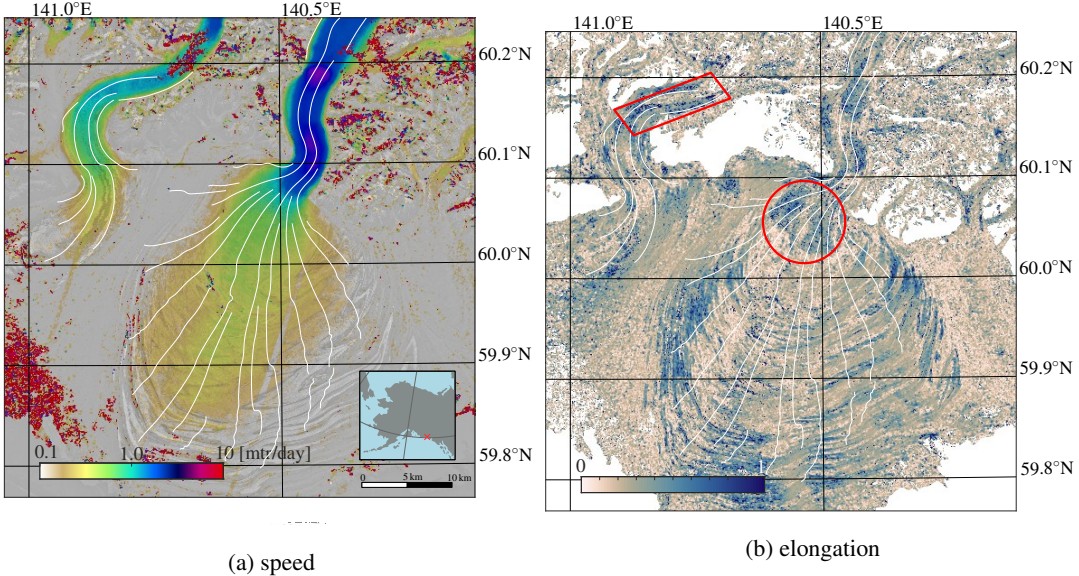

(a) speed

(b) elongation

**Figure 5.** Image displacement (Fig. 5a) and elongation of the correlation ridge (Fig. 5b).

The estimated displacements over the study region (Fig 5a) have a smooth surface. In the mountains region, small speckles
are present, as well as, a small patch on Agassiz Glacier. In this zone the transient snow line was located, so similarity is more
difficult to establish. Similarly, haze or thin cloud cover might be present at the end of the snout of Agassiz Glacier, causing
further outliers.

The pattern of elongation (Fig 5b) and shear (Fig 6a) are similar at the borders of Agassiz Glacier, as indicated by the red
parallelogram. This is not the case for many other parts, while a region which shows no extensive local shear or extension can
be seen at the start of the lobe (encircled in red). However, this region does have heavy crevassing as well (Fig. 6b). This is
not only happening in this region, but in general the Radon strength correlates well with the elongation of the ridge. Hence,
excessive shear and extension might create crevasses, and these seem to be the most dominant mechanism for asymmetrical
correlation spread. Other signals are also present in the shear estimate (Fig. 6a), but these will be highlighted later in the dis-
cussion as they are not related to correlation spread.

### 4.2.1 Orientation of crevasses and dispersion peak

The dominance of the feature orientation (Fig. 7a) to the direction of the correlation ridge (Fig. 7b) is present here, as is also
observed in the Sermeq Kujalleq case. The structure of these two independent proxies are very similar. While Sermeq Kujalleq
is dominated by clean ice, it seems also other directional features like foliations and morrain bands influence the correlation
surface.





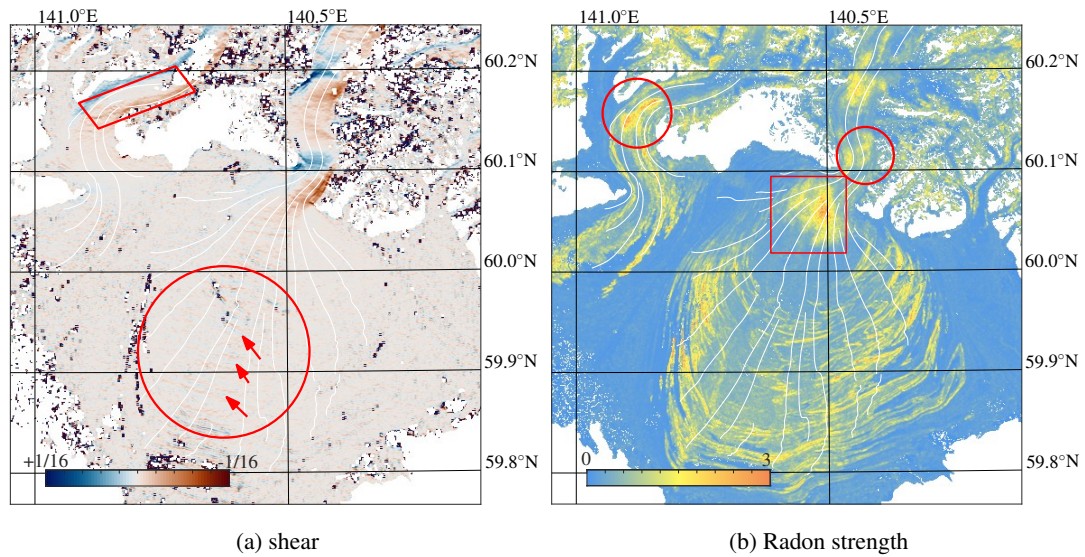

(a) shear

(b) Radon strength

**Figure 6.** Estimated surface shear, derived from the estimated velocity (Fig. 5a) and strength of image content in the form of the Radon transform.

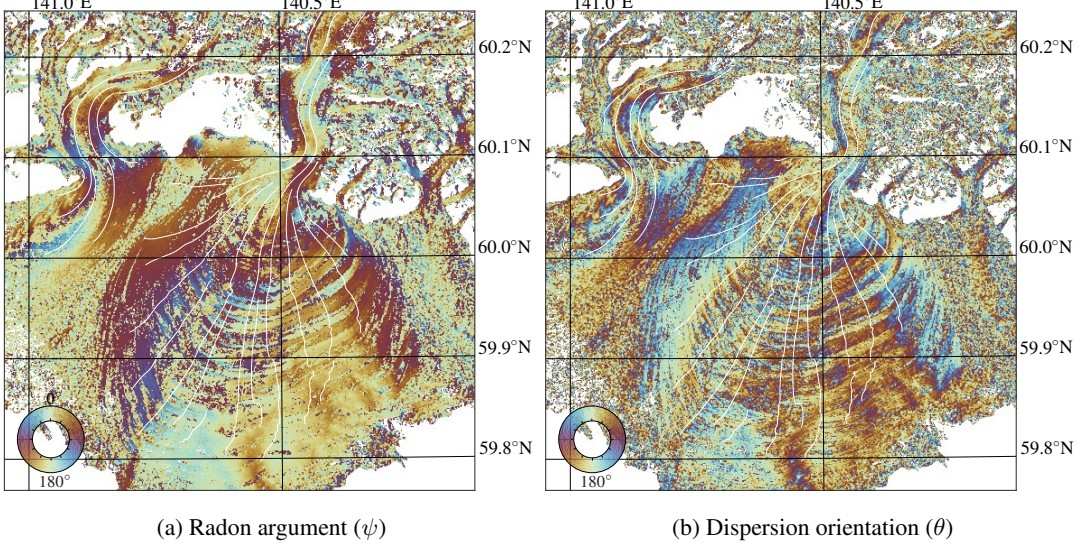

(a) Radon argument ($\psi$)

(b) Dispersion orientation ($\theta$)

**Figure 7.** Orientation descriptors over the Malaspina case study, estimated through Radon transform (Fig. 9b) and correlation spread (Fig. 9a).

## 5   Discussion

### 5.1   Interpretation of the dispersion signal

In general the main orientation of the crevasses at Sermeq Kujalleq (Fig. 3b) seems to correspond to the orientation of the Gaussian peak (Fig. 4a). When these two parameters are plotted against each other their relation becomes even more clear





(Fig. 8a). The bulk of crevasse orientations are oriented towards a North-South axis, corresponding to be perpendicular to the main flow direction. A straight correlation between both parameters is present in Fig. 8a, but do not cover the whole domain equally due to the limited distribution of flow directions. A relation with crevasse presence is profound (Fig. 8b), when the

Gaussian peak is close to symmetrical (i.e., inequality near zero) there is no clear relation, but this increases when crevasses become more apparent in the imagery through the Radon transform. The pattern of elongation of Sermeq Kujalleq (Fig. 4b) is less pronounced and does not have a clear linear relation to shear flow (not shown). A reason why no clear relation between shear flow and elongation of the correlation peak is found in our example can be due to the strong presence of crevasses that then dominate the signal and image correlation in this dataset. The dominance of crevasses in the study region could suppress

the existence a clear relation with non-uniform ice flow. Nonetheless, crevasses seem to be the dominant driver for a-symmetry in the correlation peak.

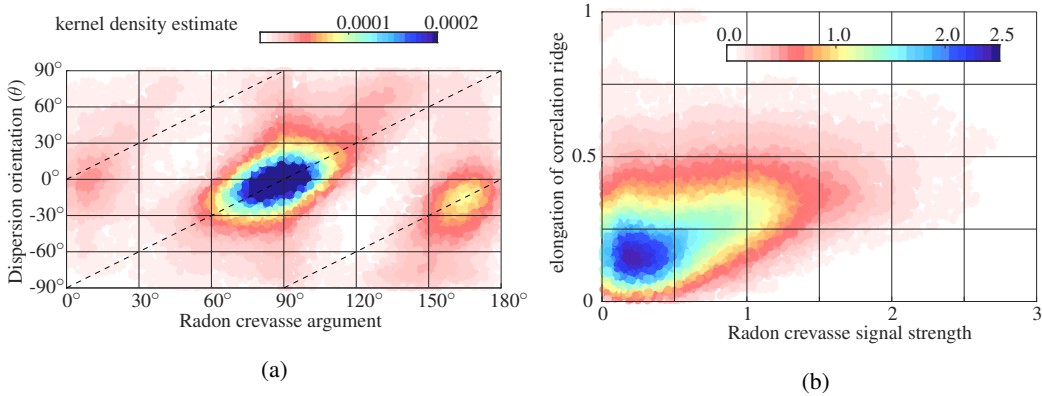

**Figure 8.** Probability density plots of results of correlation peak versus crevasse orientation (Fig. 4a & Fig. 3), and a-symmetry of the correlation peak and versus crevasse strength (Fig. 4b & Fig. 3)

## 5.2  Description of dipersion

In earlier work the handing of dispersion has been estimated through sampling statistics (standard deviation, mean absolute

difference), where displacement estimates are compared against in-situ measurements or stable terrain. The use of stable terrain for dispersion estimation has drawbacks, apart from assuming constant variance of the whole scene as mentioned earlier. Specifically, image matching is hampered by peak-locking, that favors integer displacements, thus in the configuration of stable terrain (in this case zero displacement) sample statistics will give an opportunistic estimate of precision. Individual dispersion formulations based on intensities have been proposed when least squares matching is used, but such estimates are too opti-

mistic. The large amount of pixels in a template cause the system of equations to produce a very good measurement precision, furthermore outliers in such a formulation are neglected (Maas et al., 2010). Thus the method presented here can be a direction to formulate measurement precision, without biases introduced by sample statistics and peak-locking. Another advantage of





our method is possibility to use statistical testing (Teunissen, 2000) and integration into data assimilation models or time-series construction through a full description of the co-variances (Riel et al., 2021).


We postulate that the correlation coefficient is a proxy for the confidence of a match, and are therefore less suited to function as a describtor of precision. The maximum correlation coefficient and the signal-to-noise proxy are disimilar proxies. For example, the narrow and crevassed outlet of Malaspina Glacier has low correlation scores (Fig. 9b), but a high signal-to-noise ratio (Fig. 9a). Upon closer inspection, a striking feature of multiple peaks grouped together might be observed in the

correlation score (see region inside red diamond). This pattern alligns with the sub-pixel displacement away from an integer, as the correlation score is estimated at individual steps. This off-integer bias in the correlation score can be replicated by the displacement estimate and is done so in Fig. 10b. Hence, using a correlation score as precision proxy, while it is contaminated by off-integer biases, is not recommended.

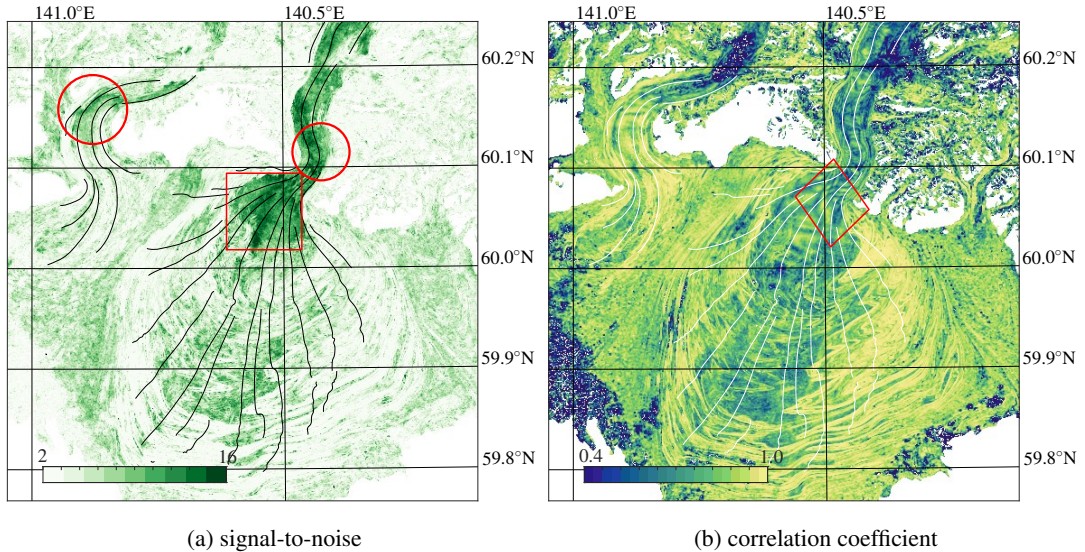

| (a) signal-to-noise | (b) correlation coefficient |

**Figure 9.** Correlation describtors over the Malaspina case study, describing signal-to-noise (Fig. 9a) and the correlation (Fig. 9b).

A second commonly used proxy for precisioin is the signal-to-noise ratio. Here we postulate that this proxy might describe

the uniqueness of a match. Very high values of signal-to-noise (Fig 9a) seem to coincide with strong crevassing (Fig 6b), as is also indicated by the red encircling. A second class of high values (see red square) is present in clean ice zones of the lobe of Malaspina Glacier, where distinct foliations occur, giving an unique fingerprint for the matching.

In this study we propose to use a Gaussian formulation to describe the matching precision. If the maximum correlation or

signal-to-noise would be a good proxy of precision, then one can expect a correlation or some form of agreement between the major axis (Fig.11) and these other proxies. However, for the data over Malaspina Glacier this does not seem to be the case as these proxies only seem to be correlated in the extreme ends. Thus, the proposed dispersion parameters do provide a new type





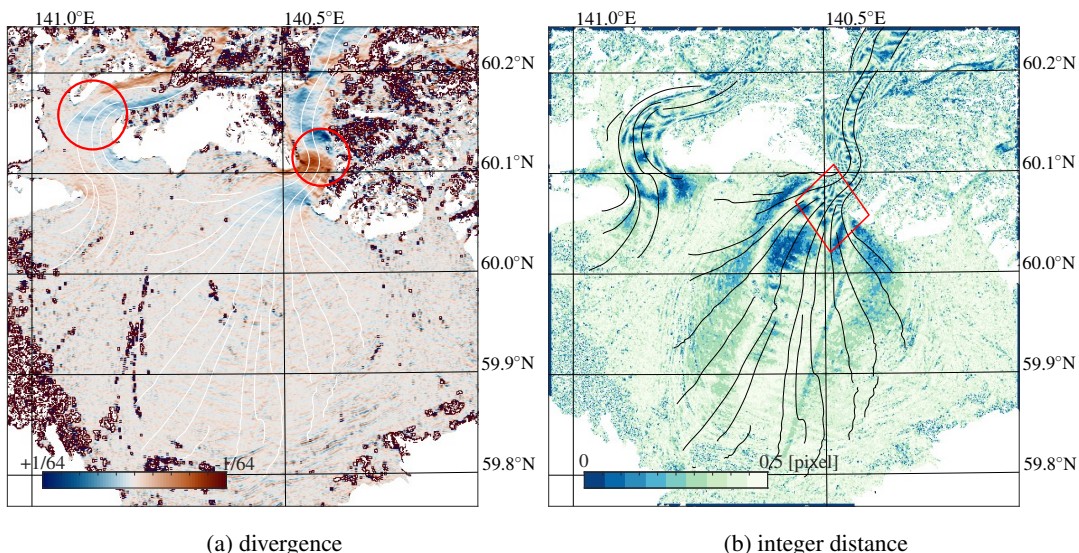

(a) divergence
(b) integer distance

**Figure 10.** Estimated surface divergence, derived from the estimated velocity (Fig. 5a) and modulus from a combination of sub-pixel displacements (Fig.D2a & D2b ) .

of data description, which we think has a straightforward connection to precision.

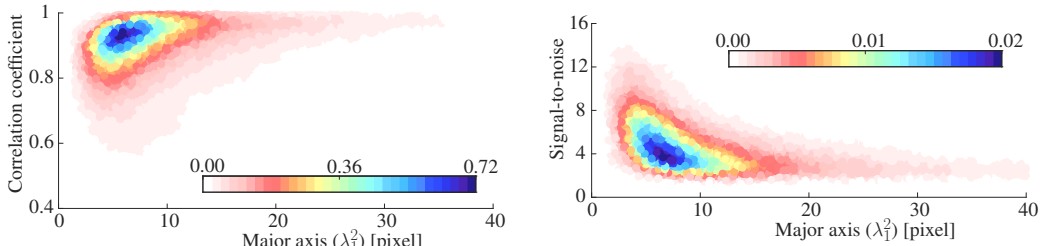

**Figure 11.** Probability scatter plots between different matching describtors.

## 5.3 Implementation issues

The implementation done here for our correlation-dispersion based method is a simple least squares adjustment, and no robust re-weighting is applied. This can result in negative variances or rank deficiency, corresponding to the white data voids in Figure 4. Causes for such anomalies can come from the logarithmic function within Equation 2, capable of transforming white to a-symmetric noise (Anthony and Granick, 2009).

In this study, the correlation computation is done in the spatial domain. Frequency domain methods produce sharp peaks, as they prescribe consistent rigid displacement at integer resolution. Furthermore, when sufficient shear occurs, or repeating





image features are present, this might result in multiple distinct but sharp peaks (Scarano, 2001). Hence, interpretation of our dispersion estimation is most suited for spatial domain methods.


Finally to demonstrate its application domain, we introduce a framework to use our dispersion estimation (see Appendix C) and resolve issues caused by missing data from neighboring displacement estimates, when estimating strain rates. This is a step towards a more integrated approach and moves away from parameter based interpolation (e.g.: (Lüttig et al., 2017)).

## 6 Conclusions

Quantifying the measurement precision of individual displacement estimates from matching repeat spaceborne images has received little attention in recent years despite the increasing efforts to produce large displacement data sets from an increasing number of suitable data. Here, we introduce a simple procedure to estimate the correlation dispersion of such displacement measurements (from either optical or SAR), through characterizing the shape of the correlation surface. We demonstrate this technique for Sermeq Kujalleq, a fast flowing and heavily crevassed outlet of the Greenland icesheet and Malaspina Glacier.

Dispersion results are compared to shear strain-rates and crevasses orientation. These results seem to indicate that crevasses are the dominant driver for a-symmetry in the correlation function. We suggest this simple procedure to estimate uncertainty of individual image matches can be useful in processing pipelines for large-volume image displacement measurements, so error-propagation can be applied on a large scale and will improve inversion of other geophysical properties. In all, we hope this demonstrates the rich information present in satellite imagery and its processing chain and might make it easier to extract

a clearer signal from such noisy remote sensing products.

*Code availability.* A simple MATLAB and Python implementation for the dispersion estimation is included in the submission. The implementation for the Radon transform can be found at runmycode.org/companion/view/2711

*Data availability.* In this study we use optical data from the Sentinel-2 satellites. Since these satellites are part of the Copernicus satellite system, which is the European Commission's earth observation programme, all data is freely available. Hence, current acquisitions can be

retrieved from https://scihub.copernicus.eu/.





## Appendix A: Schematic of an image matching pipeline

In order to clarify where in an displacment processing scheme the dispersion estimation can be implemented, a schematic of an image matching pipeline is drawn in Figure A1. It consists of the following five steps:

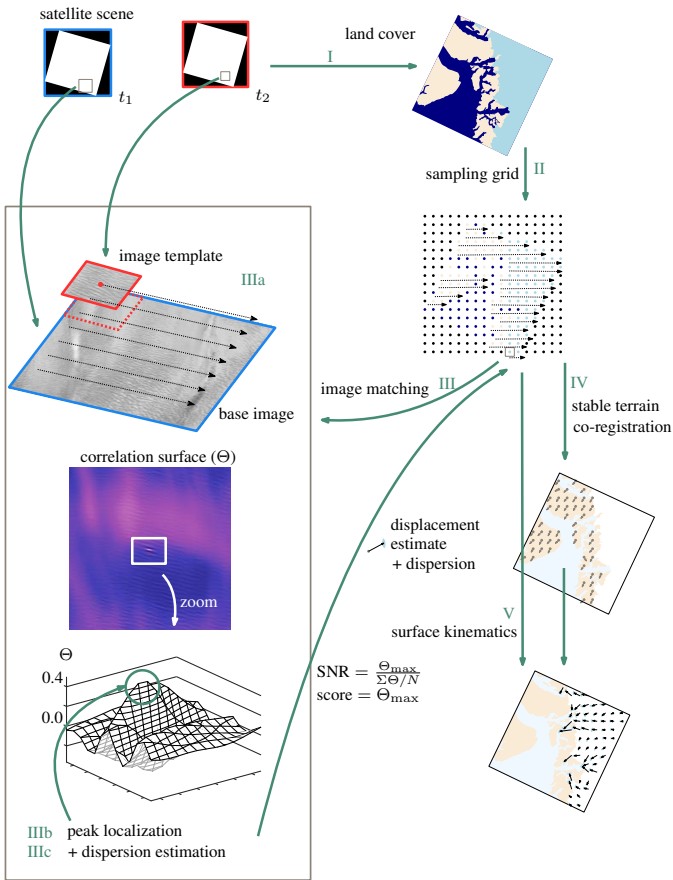

**Figure A1.** Schematic of the main procedure to generate a displacement field from a pair of remote sensing images.

I) Given the extent of the imagery, a mask is generated indicating what is ocean, land and glacier. II) A regular grid is generated, where for each location the landcover is recorded. III) For each post of the grid, a subset of the satellite imagery is used. A kernel is moved over a base image, and at every location a similarity score is estimated. This generates a correlation surface. The highest value is taken as the correct displacement. The neighboring correlation values of this peak can be used for subpixel localization, but the same values can also be used for the dispersion calculation following the method presented in this study. IV) The displacements over stable ground are used to correct offsets due to misalignment of the satellite platform. V) The co-registration parameters are subtracted from the displacement vectors, resulting in a grid of velocities and its precision.





## Appendix B: Complete derivation

A two dimensional normal distribution, with a dependency ($\rho$) in its variables can be written as (Teunissen et al., 2009),

$$I(x,y) = \frac{1}{2\pi\sigma_x\sigma_y\sqrt{1-\rho^2}} \cdot \exp\left[-\frac{1}{2\cdot(1-\rho^2)}\left(\frac{(x-x_0)^2}{\sigma_x^2} - \frac{2\rho(x-x_0)(y-y_0)}{\sigma_x\sigma_y} + \frac{(y-y_0)^2}{\sigma_y^2}\right)\right].$$

(B1)

Here $x$ and $y$ denote coordinates on two orthogonal axis, $\sigma^2$ the variance and $x_0$ and $y_0$ their mean. This formulation can be written out fully with parameters ($A, a, b \& c$) substituted for the ease of readability (Eq. 1). This results in a linear system of equations with four unknowns, so these need to be estimated through several neighboring correlation values, as written down in (Equation 2). The following operations show the transformation from one formulation to the other.

$$I(x,y) = \frac{1}{2\pi\sigma_x\sigma_y\sqrt{1-\rho^2}}$$

(B2)

$$\exp\left[\left(-\frac{1}{2\cdot(1-\rho^2)\cdot\sigma_x^2}\cdot(x-x_0)^2 + \frac{2\rho}{(1-\rho^2)\cdot\sigma_x\sigma_y}\cdot(x-x_0)(y-y_0) - \frac{1}{2\cdot(1-\rho^2)\cdot\sigma_y^2}\cdot(y-y_0)^2\right)\right]$$

$$I(x,y) = A\cdot\exp\left[a\cdot(x\text{-}x_0)^2 + b\cdot(x\text{-}x_0)\cdot(y\text{-}y_0) + c\cdot(y\text{-}y_0)^2\right]$$

(B3)

$$\ln\left[I(x,y)\right] = \ln\left[A\right]\cdot 1 + a\cdot(x\text{-}x_0)^2 + b\cdot(x\text{-}x_0)\cdot(y\text{-}y_0) + c\cdot(y\text{-}y_0)^2$$

(B4)

The substituted parameters ($A, a, b \& c$) can be written out fully as,

$$A = \frac{1}{2\pi\sigma_x\sigma_y\sqrt{1-\rho^2}}, \quad a = -\frac{1}{2\cdot(1-\rho^2)\cdot\sigma_x^2}, \quad b = \frac{2\rho}{(1-\rho^2)\cdot\sigma_x\sigma_y}, \quad c = -\frac{1}{2\cdot(1-\rho^2)\cdot\sigma_y^2}.$$

(B5)

Transferring these lumped parameters towards the Gaussian parameters (Eq. B1) is done though first formulating them in relation to the dependency ($\rho$),

$$2\rho = \frac{b}{\sqrt{a\cdot c}},$$

(B6)

$$a\cdot c = -\frac{1}{2\cdot(1-\rho^2)\cdot\sigma_x^2}\cdot-\frac{1}{2\cdot(1-\rho^2)\cdot\sigma_y^2} = \frac{1}{2^2\cdot(1-\rho^2)^2\cdot\sigma_x^2\cdot\sigma_y^2},$$

(B7)

$$\sqrt{a\cdot c} = \frac{1}{2\cdot(1-\rho^2)\cdot\sigma_x\cdot\sigma_y} = \frac{1}{2}\cdot\frac{1}{(1-\rho^2)\cdot\sigma_x\cdot\sigma_y}, \quad b = \frac{2\rho}{(1-\rho^2)\cdot\sigma_x\cdot\sigma_y}.$$

(B8)

Knowing $\rho$ makes it possible to solve the other equations and extract the variances ($\sigma_x^2, \sigma_y^2$) from the other lumped parameters (Eq. 3),

$$\sigma_x^2 = \frac{1}{-2\cdot(1-\rho^2)\cdot a} = \frac{-2\cdot(1-\rho^2)}{-2\cdot(1-\rho^2)}\cdot\sigma_x^2, \quad \sigma_y^2 = \frac{1}{-2\cdot(1-\rho^2)\cdot c} = \frac{-2\cdot(1-\rho^2)}{-2\cdot(1-\rho^2)}\cdot\sigma_y^2.$$

(B9)





## Appendix C:  Kernel computation in a genralized least squares framework

Flow describtors like strain rates can also give an insight into the geometric bedrock configuration or properties related to subglacial sliding. Strain rates can be formulated in relation to the local flow direction, giving longitudinal, transverse or shear flow, respectively. These properties are computed from velocity estimates over a close neighborhood of surrounding pixels. As strain rates are derivatives of velocities, they are particularly sensitive to the propagation of noise and errors of the input velocities. Applying thresholds and filters to the strain rates based on variations or low quality of the input velocities can lead to voids in the resulting strain rate field. Here, a methodology is introduced that is somewhat resistant to such cases caused by velocity errors or missing data.

The metodology presented here is based upon the redundancy of a kernel since it is typically formulated as a smoothed differentiation. The steps are schematically illustrated in Figure C1. When a convolution ($\otimes$) is written out as a matrix form of a displacement grid ($\mathbf{P}$) and a kernel ($\mathbf{G}$). In this matrix form one can see it clearly as a weighted linear combination from neighboring velocity measurements. For sake of clarity, the examples shown in this schematic are an implementation of two different kernels (a Sobel and Prewitt), for the two different spatial axis ($x, y$). Each collumn in the design matix ($\mathbf{A}$) is independent and is composed of positive and negative entities. The summation of all elements within the kernel need to cancel eachother out, as is indicated by the coloured elements. However, when gaps occur in the neighborhood, this energy balance is disrupted. Consequently, this lost weight should be added to others within its group, or reversely, taken away from entries with the group with an opposing sign. When the convolution is written out directly in matrix form, this allocation of energy is done by column-wise operators. If the neighborhood is out of balance, the kernel is not estimated.

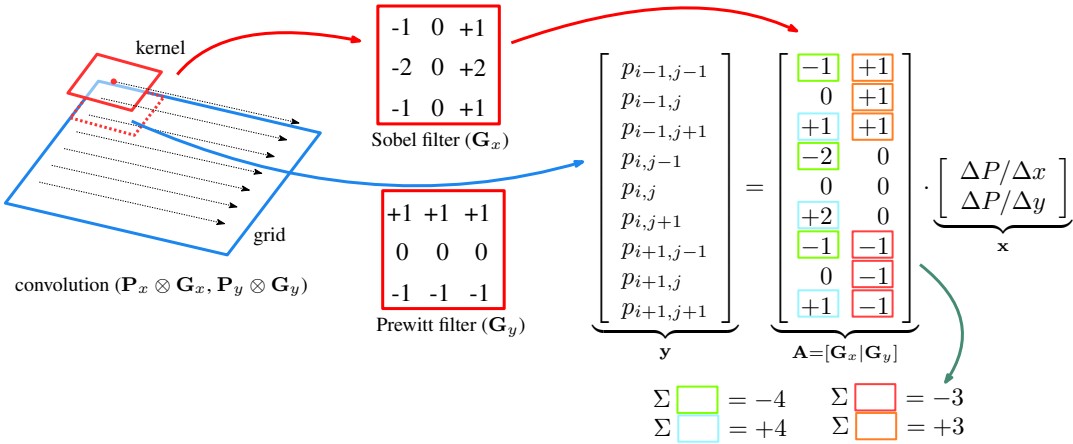

**Figure C1.** Schematic of a computation of a convolution, in this case the first derivative in vertical and horizontal direction.





In the example shown in Figure C1 the horizontal and vertical components $(x, y)$ are independent. However the dependency can also be included since formulating a convolution as a least squares estimation makes it possible to propagate the co-variances. Hence, the co-variances of the image matching as given in Eq. 4 can be used to estimate the precision and

dependencies of derived parameters, through

$$\mathbf{Q}_{\hat{x}\hat{x}} = \mathbf{A}^\top \mathbf{Q}_{yy}^{-1} \mathbf{A}. \tag{C1}$$

Hence, estimating derivatives with correct weighting, making generalized least squares possible;

$$\underline{\hat{\mathbf{x}}} = \mathbf{Q}_{\hat{x}\hat{x}}^{-1} \mathbf{A}^\top \mathbf{Q}_{yy}^{-1} \underline{\mathbf{y}}. \tag{C2}$$

Nevertheless, improvement is only made on a local level in a direct neighborhood covered by the kernel, so when large parts

are affected with regions of missing values, or the outlier detection is false, spurious fluctuations can still propegate into the final product.



## Appendix D:  additional information on the Malspina Glacier case

In this appendix additional illustrations are shown for the Malaspina Glacier, to ease interpretation of the results and to highlight
the information present in dispersion peak.

**Figure D1.** Sentinel-2 scene over Malaspina (center) and Agassiz (left) Glacier. With annotations in red to enhance interpretation.



Here we also show some sub-pixel displacement plots, as the integer (Fig. 10b) is based upon the combination of two axis, as the remainder of the modulus of displacement are shown in Figure D2. Sensor specific artifacts are present in these figures as indicated by the gray boxes or the red encircled region, see Kääb et al. (2016); Stumpf et al. (2018) for more details. They are mentioned here explicitly since these patterns are striking, and might delute the interpretation of the results in Figure 6.

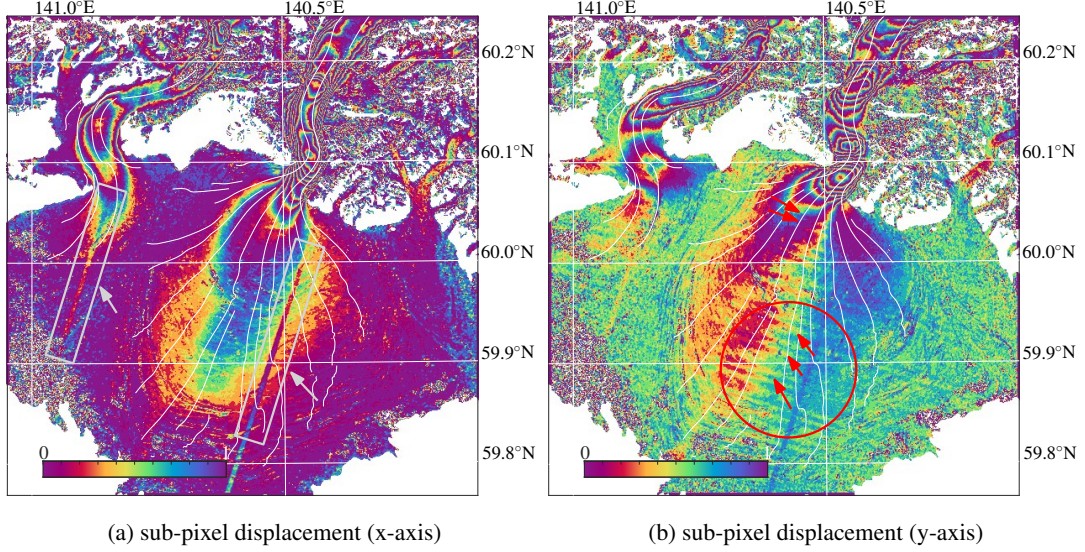

(a) sub-pixel displacement (x-axis)          (b) sub-pixel displacement (y-axis)

**Figure D2.** Rainbow colourcoded remainder of the modulus of displacement, for the horizontal and vertical direction ( Fig. D2a and Fig. D2b).





*Author contributions.* B.A. conceived the study, A.K. and B.W. contributed with comments and suggestions to the work.

*Competing interests.* The authors declare not having any competing interests. B.W. is editor of The Cryosphere.

*Acknowledgements.* This research and development has been conducted through support from Dutch research organization (NWO) via the Eratosthenes project (ALWGO.2018.044), the European Union FP7 ERC project ICEMASS (320816) and through ESA Living Planet Fellowship, Glaciers-CCI, CCI+ and EE10 HARMONY projects

(4000125560/18/I-NS, 4000109873/14/I-NB, 4000127593/19/I/NB, 4000127656/19/NL/FF/gp).



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
