# Peer review of "Correlation dispersion as a measure to better estimate uncertainty of remotely sensed glacier displacements"

_The Cryosphere, 2021_

## Author Response (AR1)

**Anonymous reviewer 1, from 03 Nov 2021**

*In "Correlation dispersion as a measure to better estimate uncertainty of remotely sensed glacier displacements", Altena et al. present an efficient and straightforward method for quantifying spatially-varying precision associated with displacement maps derived from pattern matching (pixel tracking) techniques. These techniques have become ubiquitous in glaciology and are the foundation for most contemporary velocity fields derived from remote sensing data. Here, Altena et al. utilize the topology of local cross-correlation surfaces to estimate the offset precision, which plugs in nicely to most existing workflows and should be robust for most datasets used in glaciology. Importantly, they show how these uncertainties can be linked to distinct classes of surface features, such as crevasses or shear patterns.*

*Overall, this manuscript represents an important and long overdue contribution to the field and to any researcher using remotely-sensed velocity fields. In particular, any work on data assimilation or inverse modeling absolutely needs to properly account for spatially-varying uncertainties. My comments are mostly minor/moderate and are mostly concerned with improving the presentation of the results and placing the results within the context of prior techniques and uncertainty measures.*

Thank you for your kind words.

**Summary comments**

*1) Most of the uncertainty metrics presented in the main text are those quantifying correlation surface orientation and degree of asymmetry. However, as a user of velocity fields reading a manuscript on uncertainty quantification, I did not actually see a map of velocity uncertainties for any of the study sites! I believe these maps should be front-and-center in the main text so the readers can immediately obtain an intuition on the spatial distribution of uncertainties and their relation to glacier flow speed. In my opinion, the other uncertainty metrics (e.g., correlation peak orientation and elongation) are interesting auxiliary results which should exist to enhance the presentation of the uncertainties themselves, since the latter are what most people working on data assimilation will be interested in.*

We follow this line of thinking, and the reviewer is right about this issue. We have now included the along and across flow estimates in the main figure of the results.

*2) When showing the maps of uncertainties, it would also be very useful to show how uncertainties change for different window sizes. For users wanting to adopt the methods in this manuscript, a demonstration of uncertainties for common window sizes (64x64, 128x128, or adaptive window sizes) would go a long way towards bringing awareness to window size effects on both velocity field noise levels and uncertainties.*

We think this will seriously deviate from the message, literature is full of such empirical studies. We agree with the reviewer that window size does matter (see also our response below to revierwer 2 about pproaching this as an optical flow problem). However, this is only one part of the equation, as the image content contributes as well. Hence, the results will depend on the scene and window-size used.

Nor do we want to restrict the interested reader from such exploration, therefore the processing code is available. Hence, if such subjects are of interest to the reader, the provided scripts can be used for exploration on this matter. In sum, we prefer therefore to not implement the suggestion of the referee for the sake of focus of the paper.

The provided second point of the feedback of the reviewer is well appriciated, but though interesting, the inclusion of a least-squares matching implementation (which is required for an adaptive window approach), will also deviate from the analysis given here.

*3) Pattern matching/pixel tracking has a long history in fluid mechanics (PIV, as the authors mentioned) and geosciences. For the task of deriving velocity fields over terrestrial ice, these methods have been used extensively for over*

*20 years, and several of the largest projects (e.g., MEaSUREs) also provide maps of velocity errors for their velocity products. While a brief discussion on previous uncertainty quantification methods was included in the introduction, there needs to be additional discussion and comparison of the uncertainties for methods that also go beyond the homoscedastic assumption. For example, the method used by Joughin, 2002, "Ice sheet velocity mapping..." computes the statistical offset variance for a local window centered on a given pixel. Thus, the scatter in the estimated correlation peak represents an aggregate of the different noise factors for the local window. My intuition is that the method used here (fitting the correlation surface) is a more robust approach, especially for smaller window sizes, but without a comparison of uncertainty maps for the different methods, it's hard to know for sure. Thus, related to comment (1), my suggestion is to include an uncertainty map (at least for Sermeq Kujalleq) and compare that map with a similar map from the Greenland Ice Sheet Mapping Project (GIMP).*

We understand the point made by the reviewer, but if we want to keep close to the message given by the manuscript, this does look like a strong deviation thereof. The approach presented here is an extra feature within the processing chain, which has been introduced earlier (see comments later given by this reviewer). It gives extra handholds for more advanced post-processing, such post-processing procedures that are here asked for.

The mentioned post-processing procedure of Joughin (but also many others) has a strong assumption on the homogenous consistency in flow direction and its velocity gradient. Though usefull for large outlet glaciers, it is a procedure that is further down the processing chain, from velocity pairs to mosaics. Nevertheless, we do see the benefit of such a comparison here in the response, as it might give insights on the total processing chain. Therefore, we have replicated the workflow as presented in (Joughin, 2002) and GIMP.

[Figure]

(a) vertical component [meter·day$^{-1}$].

[Figure]

(b) horizontal component [meter·day$^{-1}$].

Figure 1: Raw velocity estimates.

[Figure]

(a) vertical component [meter·day$^{-1}$].

[Figure]

(b) horizontal component [meter·day$^{-1}$].

Figure 2: Culled, filled and smoothed velocity estimates, following the workflow of (Joughin 2002) .

[Figure]

(a) vertical component.

[Figure]

(b) horizontal component.

Figure 3: Variance estimate following the workflow of (Joughin 2002), and as is given by GIMP .

Apart from being scale dependent (see provided code for specific parameter implementations), in this implementation the variance estimates seems most sensitive to shear zones. Since our method is based on the image content, it does highlighted the crevassed regions north of the outlet.

*4) Related to comment (3), estimation of pattern matching uncertainty by quantifying the topology of the cross-correlation surface has been around for a while. In the field of InSAR, software packages like the Repeat Orbit Interferometry Package (ROI_PAC) and InSAR Scientific Computing Environment (ISCE) estimate the curvature of the oversampled cross-correlation surface as an uncertainty proxy, similar to what's done here (see Rosen et al., 2004, "Updated repeat orbit interferometry package released" and the appendix of Casu et al., 2011, "Deformation Time-Series Generation . . . "). At a minimum, these should be cited in the manuscript.*

We would like to thank the reviewer for pointing this out. We have been aware of AmpCorr as it has been a subroutine used by many other processing pipelines, but the mentioning of this curvature estimate has not been mentioned explicitly.

The backbone of this package is described in the thesis of Rosen, but it distribution/accesability is less optimal. Unfortunately, the provided references do not specify how the hessian is estimated. Luckily the fortran code of AmpCorr is available, hence we have tried to translate this to Python code, so we were able to compare both methods.

[Figure]

(a) vertical component [meter·day$^{-1}$].

[Figure]

(b) horizontal component [meter·day$^{-1}$].

Figure 4: Curvature estimation as done in a similar fashion as ROI_PAC.

There is a strong agreement between both methods, crevassed regions in the center of the scene have similar patterns. However, differences are present as well. Most pronouched is the stronger differentiation between sharp or wide peaks.

[Figure]

[Figure]

(a) vertical component [meter·day$^{-1}$].

[Figure]

(b) horizontal component [meter·day$^{-1}$].

Figure 5: Dispersion estimation as proposed by (Altena et al.) .

This can be due to a smaller amount of neighboring pixels that is used for the estimation. For the Gaussian model takes a larger subset surrounding the peak. Another pattern, is the polar opposite of the estimates in the South Western side, where stable terrain is situed. Here there is a lack of structure, causing either very elongated or very narrow lobes to be estimated.

**Line-by-line Comments:**

*- Line 61: A comparison of most of these methods has been done in the field of PIV, and at least one citation should be included (e.g., see Xue et al., 2014, "Particle image velocimetry correlation signal-to-noise ratio metrics...").*

We have included this suggestion, and we agree that many algorithms and methods seems transferable. However caution should be taken, since this field is based on laboratory measurements where a controlled environment is the ussual setup. Hence, the PIV data has a clear distinction between "data-points" carrying displacement information and "background noise". For example the work by (Wieneke, 2015) is based on knowledge of the particle seeding diameter, for image matching in the wild this is less clear. The controlled environment of PIV setups makes it also easier to derive proxies or propagate errors through the whole pipeline, as in (Bhattacharya et al. 2017). Hence, the large body of work in this domain is interesting but separating particle image diameters, seed densities, displacements, and velocity gradients to come to an error budget might not be practical or easily transferable to the case presented here.

*- Section 3.1: Is any oversampling of the cross-correlation surface performed prior to fitting a 2D Gaussian? Oversampling is a common step in pattern matching and can mitigate the effects of "pixel locking" (see autoRIFT paper, Lei et al., 2021, Remote Sensing; a Gaussian pyramid upsampling scheme is used to reduce pixel locking). Additionally, it would likely provide more data points for the 2D Gaussian fit. Without oversampling, wouldn't there be situations where the cross-correlation surface is highly concentrated at a single location (i.e., a high SNR case), in which case the 2D Gaussian fit would be poorly conditioned.*

Since spatial domain image matching is performed, no oversampling is applied. Oversampling is a strategy to supress peak-locking when frequency domain methods are applied (see comment for line 257).

Looking at the results at the Figure 4 and 5 above, the indication given by the reviewer seems to go towards the opposite direction in this case. It seems the parameter estimation through an Hessian, using 5, 7 or 9 data points, makes the estimation too optimisitic or exegurate the lobe. Using 25 data points does seem to benefit the estimation, but if the window size shrinks or the image content changes to speckles, this might not be the case. To a degree interpolation is present in the procedure, since a weighting function is optionally implemented in the code.

*Also, what is a typical zoom window size (centered around the peak) for fitting the Gaussian? I assume the entire cross-correlation surface is not used for the fit. If the zoom window size encompasses multiple peaks (primary + secondary peaks), how is that situation handled in the processing chain?*

This crucial information was not included in the text, but now: "The direct neighborhood is used here (radius=1) when the peak next to the border, otherwise a two pixel radius is used. " is included.

In theory the second peak might be present in this outershell, and would corrupt the estimation. One strategy is a distance weighting, as mentioned above. A second strategy is to account for the corruption through data-snooping, but we did not experienced such defects, and this is not implemented, partly because the exponential function corrupts the noise distribution.

*- Line 119: I think care should be taken when referring to general covariance matrices. In general, off-diagonal elements describe dependences between variables. The temporal/spatial relational dependencies mentioned on this line are for a completely different set of variables.*

We think it should be mentioned here, to give an outlook where this can go to. Seen from a bundle block adjustment perspective, it is logical that if a network of imagery are matched, the depedencies should be included. A sparse matrix is computationally more friendly, but formulations with off-diagonal elements might make it easier to identify outliers and biases.

*- Figure 2: This figure could likely go into Appendix B since, by itself, it doesn't add too much to the discussion.*

Has been moved to the appendix.

*- Line 152: It's a bit incomplete to say that co-registration is not applied to the image pair beforehand. Nominally, users will use the image metadata to approximately co-register the images in order to reduce the need for large search windows. Are the authors referring to refinement of image registration over stable ground? If so, that should be stated more clearly.*

Registration of Sentinel-2 is currently done through its sensor recordings, though current data does use the Ground Reference Image (GRI). Legacy imagery does fluctuate within an absolute registration error bound of 10 meter (Kääb et al. 2016). We have now included the mentioning of orthrorectification, though the Level-1A data is not available. The suggestion by the reviewer is a bit confusing, as we follow the terminology of (Nuth & Kääb), since neither Sentinel-2 scenes are ground truth, and a relative off-set/ mapping function is not estimated.

*- Line 163-164: What does "extensive flow" mean? Extensive extensional strain?.*

Has been adjusted.

*- Figure 3/4: I suggest moving the Radon crevasse orientation in Figure 3 to Figure 4 to better compare with the correlation peak orientation..*

The figure has been moved.

*- Figure 8 caption: Please also include that these results are for Sermeq Kujallec..*

The glacier name is now included.

*- Line 212: Doesn't strong shear flow generally result in crevasse formation? It seems a little odd to categorize these features into two distinct classes.*

This is true, but crevasses are transported, where they still can have an effect on the correlation spread.

*- Line 221-222: If I'm not mistaken, pixel/peak locking is a consequence of estimating the center of mass of a few discrete pixels. Methods that fit the correlation surface with a model (as is done here) should thus avoid those issues, right? Can't peak locking also be mitigated by oversampling of the cross-correlation surface? (see my comment above*

*about oversampling of the cross-correlation surface).*

Peak-locking issues arise distinctively when frequency based methods are used and this issue is raised as this bias is systematically favoring lower error statistics. The localization of this peak location is not analzed here, neither does the dispersion estimation compensate for such biases.

Oversampling might work, another methods to compensate for this peak-locking bias is shifting by .5 pixels.

*- Line 224: Please add a few words on what least squares matching is. What do "intensities" refer to here?*

We have limited ourselves to a brief description of template matching within a least-squares framework. Hence, the paper by (Förstner, 1987) roots to this concept and connects to the work by (Maas et al.). We find it out of scope for this paper to describe template matching with an affine model, which is typically done in a least squares framework as well, and is typically known as least squares matching.

*- Line 232: describtor change to descriptor*

Corrected.

*- Line 234-238: These sentences are a bit confusing to me. I don't quite understand how the sub-pixel displacements influence the correlation score. Is it because a correlation peak's energy becomes evenly distributed across multiple pixels in the cross-correlation surface? If so, this seems similar to the pixel locking effect and could perhaps be mitigated by oversampling of the cross-correlation surface (see my earlier comment).*

Here we have used spatial domain matching, hence we do not expect peak-locking to be very much prenounched. Peak-locking occurs due to sharp sidelobes, as pure displacement through frequency methods resemble a sinc-function. Thus correlation scores next to the maximum can have negative values. This will influence the location estimate if a gaussian or polynomial model is used. Hence, we do not think this pattern is peak locking, and this is only an observation showing the absolute correlation value is influenced by the underlying sampling.

*- Line 239: precisioin change to precision*

Corrected.

*- Line 247 and Figure 11: Actually, to my eye, it seems like the signal-to-noise ratio and major axis have a reasonable correlation, e.g. high SNR is inversely proportional to major axis. This would make sense as this means the cross-correlation peak is more concentrated and compact relative to the noise floor.*

We agree, but this is only true for the extreme cases, the bulk of the matching is lumped together.

*Again, a map of total uncertainty (or even just major axis) would be illuminating when compared to the maps of SNR and correlation coefficient.*

The suggested metrics are shown in figure 6. Though the relation suggested by the reviewer can not be seen by us.

[Figure]

(a) peak to noise of the correlation score.

[Figure]

(b) absolute value of the maximum score in the correlation surface.

Figure 6: Score values for the Greenland case.

*- Line 257: How do frequency domain methods prescribe displacement at integer resolutions? It's well known that a real-valued shift between two signals in the time/spatial domain will lead to a ramp in the frequency domain. This ramp can be estimated to achieve sub-pixel resolution (Leprince et al., 2007).*

Here as well as mentioned in the study, correlation is calculated in the Fourier space, and transformed back to the spatial domain. Hence, a correlation surface is the result. Since the inverse Fourier transformation is a discrete form, smearing and peak-locking is happening.

In a first step (Leprince et al. 2007) do this procedure, to get the integer displacement. Then the window gets adjusted/refined to the displaced location, where again correlation is done in the Fourier-space. However, now there is no transfromation back to the spatial domain, and a plane is fitted. The fitting of a plane/ramp overcomes the drawbacks of the inverse discrete fourier transformation.

*- Line 275: It would probably be useful to specify "physical signal" if one is referring to improvement of data assimilation/inverse methods.*
Included.

**Anonymous reviewer 2, from 07 Dec 2021**

General Comments

*This paper conducts an excellent study into how the uncertainty of photogrammetry-based ice diplacements varies over the velocity field, a neglected subject in glaciological studies. However, I am left wonddering exactly how much the proposed method improves on existing isotropic methods. Line 34: The hypothesis of this paper is stated in line 34-35: "IIn our opinion the assumption of constant variance (homoscedasticity) does not hold, as displacement extraction is based upon pattern matching of small subsets of imagery, where the image content influences the displacement precision. Is this demonstrated? I did not find an explicit answer. I am left wondering the degree to which the assumption of homoscedacity is violated and if there are glaciological settings in which this assumption is appropriate or acceptable.*

There are many ways how to convey this principle, we have opted for a practical explanation. In the text we have described the phenomena based on image structure, a second way is by demonstrating examples as is done in Fig.1 of the manuscript. However, a more theoretical top-down approach is also possible;

The issue of homoscedasticity can also be approached from a perspective of optical flow. Pattern matching and optical flow can be seen as interchangable techniques, as they are mathematically identical (Lemmens, 1988). If image gradients are present in several directions, the span of the matrix is sufficiently large. However, when there is a predominant direction in the image gradients, the optical estimation becomes ill-posed (also known as the aperture problem). Hence, treating displacement axis indepedent does not hold nor is a fixed precision term sufficient.

This is now included in the text. To pull this property out even further we have implemented an optical flow estimation as well. In this way the single values of such estimations can be used to see to what extent these differ. The single values for our dataset are shown in figure 7.

[Figure]

[Figure]

(a) first single value.       (b) second single value.

Figure 7: Single value estimates.

Distinct difference between both values, indicate homoscedasticity is not present within this dataset.

**Specific comments**

*Line 5 and 13: please state how the correlation peak is related to velocity uncertainty. It would also be helpful to the reader if you could explain what a dispersion estimate means in this context or how it relates to uncertainty.*
Additional explanation is included, but kept to the minimum since it is in the abstract.

*Line 19: change "automatic constructing" to "automatic construction".*
adjusted

*Line 47: Insert comma after "In this contribution"*
inserted

*Line 49: remove "then for instance"*
removed

*Line 61: change "also other metrics can " to other metrics can also"*
changed

*Line 68: " A lot" is a bit informal. Maybe change to significantly or greatly?*
changed

*Line 69: insert comma after "For example".*
done

*Line 74: change "it are these" to "it is these"*
changed

*Line 88: Insert comma after " and in particular its peak"*
done

*Line 95: insert comma after "is perceived as a probability density function"*
done

*Line 105: Make "a detailed derivation thereof..." its own sentence.*
is now separated

*Line 113: Insert comma before " and correlation ridges with different"*
done

*Line 124: Insert "the" between "from" and "standard error axis". Remove comma before "to a description of standard error ellipse..."*
implemented

*Line 132: Change "wholes" to holes*
deleted

*Line 176: Remove comma following Maslaspina Glacier*
done

*Figures 5 and 6: please explain the significance of the regions outlined in red. It would be helpful to the reader if other significant aspects of this figure were described in greater detail in the caption.*
These are now included.
*Figure 9: Please check spelling, describe what is significant about regions outlined in red. It would be helpful to fill out the figure caption more and describe what is significant or of interest in this figure.*
Done.

*Line 249: Here and elsewhere, terms such as "we think" and "opinion" are used. Have these opinions been validated in this study? It is not clear to me. If yes, this should be stated explicitly. Otherwise, it will be difficult for the reader to know what to do with the results of this study.*

Emphasis on dispersion within the literature has been very limited, while with this study we have tried to demonstrate its usfullness. Furthermore, we have tried to find the mechanisms behind the spread, based upon metrics from the same dataset. Two different glacial systems were choosen, but they do not span the whole range. Hence, we are cautious about our interpretation. The first reviewer also asked for additional results to test some hypothesis (see above), but we hope we have been able to show that these patterns are not so clearly extractable.

We see an article as a step towards a better understanding of such remote sensing methodologies. We found corresponding patterns and relations, and some of these can be isolated, but more analysis are possible. The code for out methodology is given, so others have the opportunity to build on top of our effort.

*Line 274-275: It would be helpful for the reader if you could briefly discuss what would be needed to "extract a clearer signal from noisy remote sensing products"*

We have adjusted the sentince, and this line relates to the former sentince.

---

## Author Response (AR2)

Institute for Marine and Atmospheric research Utrecht University, the Netherlands

April 28, 2022

Editorial office of The cryosphere Bahnhofsallee 1e 37081 Göttingen, Germany

Dear dr. Kang Yang:

Attached you find an improved manuscipt, which has your suggestions implemented of the second round of review.

We hope you find the corrections sufficient to start the publishing process.

Thank you for your time and consideration. Kind regards,

Bas Altena, Andreas Kääb & Bert Wouters

encl: response to Reviewer 1

**Anonymous reviewer 1, from 22 Feb 2022**

Overall, the authors have done a good job addressing reviewer concerns. I especially commend the authors for going through the trouble of implementing the AmpCorr algorithm and the workflow of Joughin, 2002 for comparing the different uncertainty metrics. This is no easy task, and hopefully this more detailed comparison is illuminating for readers who are willing to get into the weeds. I only have a few minor comments remaining for typos and clarifying points (note: line numbers are referenced to the revised manuscript).

- *Line 103: "directly relate the correlation peak"*  $\rightarrow$  "*directly model the correlation peak"*. sentince has been corrected.

- Figure 2: The caption and the title for maps (b) and (c) indicate variance, but units of mday are used. Do the authors mean standard deviation instead? Or are the units  $(mday)^2$ ?

This is correct, but standard deviation ( $\sigma$ ) instead of variance ( $\sigma^2$ ) is given, hence this is now adjusted in the figure.

- Figure 2: As mentioned in the response when comparing to ROI\_PAC, the south-western side has high uncertainties, and the authors mentioned "lack of structure", but the ROI\_PAC uncertainties are low, which seems inconsistent to me, but I'm sure there's an explanation for this. The authors alluded to the explanation in Line 179 ("where crevasses occur"), but didn't state explicitly the orientation of the crevasses relative to the flow direction. I would suggest a few short words on this for the readers' benefit.

Concerning the first point, an intermediate step in our argumentation is not given, which might have caused the difference between both estimates; As there is a lack of structure results a low singal to noise ratio (SNR) is present. Instead of a single pronounced peak in the correlation function, a noisy landscape will be present. This has an effect on the fitting, especially for our proposed method, as it uses more of the surrounding than the ROI\_PAC method.

Concerning the second point, we have now more clearly stated more clearly what kind of crevassing occurs: i.e.: sheared.

- *Line 179: "south-eastern part". Do you mean south-western?* This suggestion has been implemented.

- Figure 3: Might want to place a white box behind the Greek symbols on the map for easier viewing. A white background is now included in the figure.

- Line 242: From my original comment and the authors' response, it seems like we have slightly different definitions of peak-locking. From what I can find in the literature (PIV and stereo vision), peak-locking is not unique to frequency domain methods and can be present in discrete methods in the spatial domain as well. Therefore, if the authors have a reference for peak-locking as it pertains to frequency domain matching, please include it here. A reference is included in the new manuscript.

It is a very interesting aspect that the reviewer is rasing, but if no references are given it is a bit difficult to respond.

Nevertheless, to us it is not a surprise that the effect of peak-locking might be present in spatial methods. The underlying cause might be rooted from the same origin, since many if not all spatial correlation methods use the discrete Fourier transform under the hood. This speeds-up the convolution, but as a by effect might introduce a integer preference.

- Line 246: I would merge the two sentences, e.g. "...such estimates seem too optimistic"  $\rightarrow$  "such estimates can be highly influenced by sample statistics where a large amount of pixels in a template cause...""

**Sentences are merged now.**

- Line 278: Frequency domain methods produce sharp peaks where? In the correlation surface that's transformed back into the spatial domain? Please specify here. It is now better specified in the text.

Also, as the authors mentioned, spatial domain methods that do not take into account affine transformations within the image template also assume rigid translation, so there's a distinction there that should be briefly mentioned here.

Theoretically, this might be the case, but in practice spatial domain methods are less sensitive (see Figure 4.10 in [Altena 2018]). While Fourier methods have a distinct separation into a phase and amplitude domain (see e.g.: [Altena and Wouters, 2021]). An eloboration into this domain might not be of interest to the reader.

- *Line 346: "metodology"*  $\rightarrow$  *"methodology"* Correction is done.

**References**

[Altena 2018] Observing change in glacier flow by using optical satellites

[Altena and Wouters, 2021] Shadow cast tracking for deduction of elevation data through affine matching methods on optical satellite imagery